# Leucine-Rich Glioma-Inactivated 1 (LGI1) Protein Stimulates Proliferation and IL-10 Production in Peripheral Blood Mononuclear Cells of Patients with LGI1 Antibody-Mediated Autoimmune Encephalitis In Vitro

**DOI:** 10.3390/ijms25052581

**Published:** 2024-02-23

**Authors:** Alexander Goihl, Dirk Reinhold, Annegret Reinhold, Burkhart Schraven, Peter Körtvelyessy

**Affiliations:** 1Institute of Molecular and Clinical Immunology, Otto-von-Guericke-University Magdeburg, 39120 Magdeburg, Germany; alexander.goihl@med.ovgu.de (A.G.); dirk.reinhold@med.ovgu.de (D.R.); annegret.reinhold@med.ovgu.de (A.R.); burkhart.schraven@med.ovgu.de (B.S.); 2Health Campus Immunology, Infection and Inflammation (GC-I3), Medical Faculty, Otto-von-Guericke-University Magdeburg, 39120 Magdeburg, Germany; 3ChaMP, Center for Health and Medical Prevention, Otto-von-Guericke-University, 39120 Magdeburg, Germany; 4German Center for Neurodegenerative Diseases (DZNE) Magdeburg, 39120 Magdeburg, Germany; 5Klinik und Hochschulambulanz für Neurologie, Charité—Universitätsmedizin Berlin, Corporate Member of Freie Universität Berlin and Humboldt Universität zu Berlin, 10117 Berlin, Germany; 6Labor Berlin, Innovations, Sylter Strasse 2, 13353 Berlin, Germany

**Keywords:** LGI1 antibody-mediated LE, T cell proliferation, cytokine production, IL-10, autoimmune encephalitis

## Abstract

Limbic encephalitis (LE) due to anti-leucine-rich glioma-inactivated 1 (LGI1) antibodies is an autoimmune disease characterized by distinct clinical features unique to LGI1 LE, such as faciobrachial dystonic seizures. However, it is unclear whether an additional disease-related LGI1 antigen-specific T cell response is involved in the pathogenesis of this disease. To address this question, we studied the effect of recombinant LGI1 on the proliferation and effector-specific cytokine production (IFN-γ, IL-5, IL-10, and IL-17) of peripheral blood mononuclear cells (PBMCs) from patients with LGI1 LE and healthy controls. We observed that recombinant LGI1 stimulated the proliferation of PBMCs from patients with LGI1 LE, but not from healthy controls. Cytokine measurement of cell culture supernatants from PBMCs incubated with recombinant LGI1 revealed a highly significant increase in IL-10 release in PBMCs from patients with LGI1 LE in comparison with healthy controls. These results suggest that LGI1-mediated stimulation of PBMCs from patients with LGI1 LE leads to the establishment of an IL-10-dominated immunosuppressive cytokine milieu, which may inhibit Th1 differentiation and support B cell proliferation, IgG production, and IgG subclass switching.

## 1. Introduction

Autoimmune encephalitis (AE) caused by antibodies (abs) against the leucine-rich glioma-inactivated 1 (LGI1) protein has a common clinical phenotype and disease progression most of the time, starting from faciobrachial dystonic seizures and/or epileptic seizures, leading to full-blown limbic encephalitis (LE) with permanent cognitive impairment and hippocampal atrophy without instant immunosuppressive treatment [1,2,3]. LGI1 abs from these patients elicit pathological electrophysiological and cellular effects in mouse models [4,5,6,7]. LGI1 abs are not always detected in the CSF of these patients, leaving this diagnostic feature to be of limited use in contrast with other AEs, and pointing at a possibly different pathomechanism in this AE [8,9,10]. On the other hand, the existence of plasma cells producing LGI1 abs was demonstrated in the CSF of patients with LGI1 AE [9,11]. This notion received further support when identifying that LGI1 AE might be an immunoglobulin G4 (IgG4)-mediated disease characterized by a lack of a significant neuroimmunological response, as measured in the CSF and serum. IgG4-related diseases, such as muscle-specific kinase (MuSK)-antibody-mediated myasthenia gravis or Mikulicz’s disease, are a specific class of immunological diseases with a predominant role of IgG4 antibodies in their pathomechanisms. These IgG class 4 antibodies are poor complement binders and are different in their clinical picture from other IgG-related/mediated diseases, such as other autoimmune encephalitis. These findings are consistent with other IgG4-mediated neurological diseases [8,12]. In contrast, IgG1-mediated AEs, such as AE with antibodies against N-methyl-D-aspartate receptors (NMDARs), show distinct and elevated neuroimmunological reactions, well in line with the proposed antibody-mediated pathomechanisms and clinical signs [13,14]. The contradictory results on the presence of specific plasma cells ex vivo and the low predominance of abs in vivo in LGI1 AE open up the field for an additional involvement of other immunological pathomechanisms besides the direct ab-mediated effects in this disease.

However, it is unclear whether a disease-related LGI1 antigen-specific effector T cell response occurs in this disease with subsequent cytokine release by disease-related, antigen-specific immune cells, which is involved in the pathogenesis of LGI1 LE. To address this question, we studied the antigen-specific effect of recombinant LGI1 on the proliferation and cytokine production of peripheral blood mononuclear cells (PBMCs) from patients with LGI1 ab-mediated LE and healthy controls.

## 2. Results

### 2.1. LGI1 Protein Stimulates Proliferation of PBMCs from Patients with LGI1 ab-Mediated AE

To address the question of whether a specific T cell response to the LGI1 antigen with subsequent cytokine release occurs in LGI1 ab-mediated AE, herein, we evaluated the effect of recombinant LGI1 on the proliferation and cytokine production of PBMCs from 4 patients with LGI1 ab-mediated LE and 10 healthy controls in vitro. PBMCs were incubated with human, recombinant native, and denatured LGI1, as well as with a control supernatant of non-transfected HEK 293 cells or a medium, respectively.

Analysis of the cell cultures was conducted in a label-free proliferation assay using an IncuCyte S3^®^ live cell analysis system. We observed that native and denatured LGI1 proteins stimulated the proliferation of PBMCs from three of four patients with LGI1 ab-mediated LE, but not those from healthy control volunteers, presenting as an increase in the cell count per image as well as the stimulation index (Figure 1A,B). The cell proliferation was not changed in cultures of PBMCs from patients and healthy controls containing the medium alone or the supernatant of non-transfected HEK 293 cells. Representative pictures of the PBMC cultures are shown in Appendix A. The mitogen phytohemagglutinin (PHA) used in parallel cell cultures was capable of stimulating the proliferation of PBMCs from all patients and healthy volunteers.

### 2.2. LGI1 protein Stimulates IL-10 Production in PBMCs from Patients with LGI1 ab-Mediated AE

We further investigated the LGI1 antigen-specific cytokine production of PBMCs incubated with LGI1 for 4 days. Concentrations of IFN-γ, IL-5, IL-17, and IL-10 were measured in cell culture supernatants by specific ELISA systems. The production of IFN-γ as the Th1 cytokine, IL-5 (Th2 response), and IL-17 (Th17 response) was not significantly induced in cultures of PBMCs after stimulation with native or denatured LGI1, neither in the cultures of healthy control volunteers nor of patients with LGI1 ab-mediated LE (Figure 2).

Notably, the production of the multifunctional immunosuppressive cytokine IL-10 was found to be significantly (*p* < 0.01) increased in the supernatants of PBMCs from all four patients suffering from LGI1 ab-mediated LE after stimulation with native LGI1 (1459 ± 475.9 pg/mL) in comparison with the supernatants of PBMCs from healthy controls (Figure 2; Appendix A). In the supernatants of PBMCs from both patients and healthy control volunteers incubated with denatured LGI1, only small amounts of IL-10 could be detected.

## 3. Discussion

We observed that recombinant LGI1 protein stimulated the proliferation of PBMCs from patients with LGI1 ab-mediated LE in vitro, but not of PBMCs from healthy control volunteers. Since a B cell response is often triggered by a T cell response, we therefore started to examine the T cell response in autoimmune encephalitis. The results demonstrate the possible existence of autoreactive pathogenic T cells in these patients, which has not been reported previously in any known AE so far.

PBMCs from LGI1 ab-mediated LE patients produced significantly increased concentrations of the multifunctional pleiotropic cytokine IL-10 after stimulation with native, recombinant LGI1 in vitro. It should be noted that PBMCs are a cell mixture of T cells, B cells, NK cells, monocytes, and dendritic cells, which all produce cytokines. In the present experimental setting, we cannot clearly say which cells produce which amount of IL-10. It is undisputed that IL-10 is also produced by T cells following such stimulation with the LGI1 antigen.

Alongside TGF-ß, IL-10 is one of the most potent immunosuppressive factors, which inhibits T-cell activation in particular. In myeloid cells, such as monocytes, dendritic cells, and macrophages, IL-10 inhibits cell activation, leading to reduced secretion of proinflammatory cytokines, and diminished adhesion and expression of co-stimulatory molecules, thus preventing T cell activation.

In contrast, IL-10 promotes B cell activation and antibody production, including a class switch to IgG4. Regarding B cells, IL-10 can also suppress B cells in certain pathological conditions [15].

Thus, IL-10 production is also an indicator of a shift in the immune response toward Th2 concomitant with a class switch to IgG4 [16,17]. Since LGI1 ab-mediated LE is most likely an IgG4-related disease and differentiation into this IgG4 subclass can be promoted by IL-10, these data are well in line with the literature. There has not been a lot of research conducted involving IL-10 and neurological diseases but, in general, it looks like IL-10 should be more of a focus in IgG4-related diseases.

Nevertheless, increased IL-10 may cause additional effects. It was shown that IL-10 levels promote the survival of neurons and could, therefore, support a delayed disease progression [18,19]. The precise effect of IL-10 seems to be determined by the underlying disease [15]. Future studies are needed to understand this IL-10 response in LGI1 ab-mediated AE.

IL-5 induces the development, release, and attraction of eosinophils and, moreover, promotes IgA synthesis, which is not consistent with the known LGI1 LE pathomechanisms [20]. No significant changes in IL-5 and IFN-γ levels have been reported in the CSF of LGI1 patients [10,12,21]. This raises the question of whether these cytokines are produced and inactivated directly at the site of inflammation. Another possible reason could be that due to low cell numbers, the cytokine production is too low to be measured in the CSF. Due to the rarity of LGI1 ab-mediated LE, patient numbers are low, leading to the limitation of the small number of patients included. PBMCs did not produce IL-17 in the presence of recombinant LGI1. These facts may point to a reduction in, or absence of, a Th17 response. The low IL-17 response was surprising since some groups have already stated the presence of IL-17 at least in the CSF of LGI1 LE, but others have not [12,22].

The limitations of this study are partly due to the rareness of the disease alongside having a small sample size and heterogenous baselines per patient, on the one hand, and a time-demanding setup preventing a multi-center study on the other. Due to the limited amounts of PBMCs from patients with LGI1 ab-mediated LE, we quantified the cytokine concentrations in PBMC culture supernatants only on day 4 of the experiment. In future studies, this should be performed at several time points. Moreover, another limitation of the present study is that the stimulation experiments with LGI1 were not additionally carried out on purified T cells, e.g., with irradiated feeder cells as antigen-presenting cells.

## 4. Materials and Methods

### 4.1. Production and Isolation of Recombinant Human LGI1

HEK 293 cells (#CRL-1573, ATCC, Manassas, VA, USA) were stably transfected with the human LGI1 gene containing the pCMV3-SP-N-His vector (accession no. NM_005097-3, Sino Biological, Eschborn, Germany) according to the calcium phosphate protocol established by Wigler et al. [23]. The success of LGI1-containing plasmid transfer was assessed 48 h after transfection by sodium dodecyl sulfate–polyacrylamide gel electrophoresis (SDS-PAGE) and Western blotting using harvested supernatants with the secreted LGI1. An amount of 1 µg/mL of the polyclonal rabbit anti-human LGI1 ab (#ABIN720430; antibodies-online GmbH, Aachen, Germany) and, subsequently, the HRP-conjugated donkey anti-rabbit F(ab’)2 (#711-036-152; 1:10,000; Dianova, Hamburg, Germany) were used for the detection of the recombinantly expressed protein. LGI1 was purified from the collected supernatants by Ni-chelate chromatography. The protein integrity was verified by SDS-PAGE/Western blot, as mentioned above, and the purity was confirmed by SDS-PAGE with Coomassie staining (see Appendix A). Denaturation of the LGI1 protein was performed in 8 M of urea with 10 mM of dithiothreitol at 60 °C for 1 h with an ensuing rebuffering in PBS.

### 4.2. Patients and Healthy Control Volunteer Characterization

Four patients suffering from LGI1 ab-mediated AE (3 females and 1 male, aged 46 to 67 years) were identified at the Department of Neurology of the University Hospital Magdeburg. The control group consisted of 10 age- and gender-matched healthy volunteers (7 females and 3 males) with an age ranging from 31 to 63 years (Appendix A). Every patient and healthy control volunteer gave written and informed consent. This study was approved by the local Ethics Committee (study number 100/16, approved in 2016 by the Ethics Committee of the University Hospital Magdeburg). Every blood sample was taken in 2019, except patient Nr.4, whose blood sample was taken in 2020.

Patient 1: This patient was admitted in 2015 with four consecutive grand-mal seizures on one day without a known history of epilepsy. Retrospectively, cognitive decline had been occurring for 6 months. Serum analysis showed an LGI1 titer of 1:100, and MRI showed a FLAIR-intense signal. Treatment was started with methylprednisolone intravenously and then switched to plasmapheresis and 1000 mg of rituximab, twice.

Patient 2: This patient had faciobrachial seizures of up to 100/day. The serum LGI1 titer was 1:160, and the MRI was normal. After two treatment cycles with methylprednisolone, this patient was symptom-free.

Patient 3: This patient presented with complex partial seizures and up to 50 faciobrachial seizures. The serum LGI1 titer was 1:1000, and the MRI was normal. Treatment was started with methylprednisolone intravenously and then switched to plasmapheresis and 1000 mg of rituximab, twice.

Patient 4: After a 2-year-long period of cognitive decline, complex partial seizures started. The serum LGI1 titer was 1:20 at admission. The MRI showed a picture consistent with a post-LE condition with a bilateral left > right hippocampal atrophy. Treatment was started with methylprednisolone intravenously and then switched to plasmapheresis and 1000 mg of rituximab, twice.

More clinical and immunological data on these 4 LGI1 patients have also been published in Ludewig, Salzburger, by Goihl et al. [24]. Furthermore, every cytokine level per patient is shown in the Appendix A.

### 4.3. Cells

Human PBMCs were isolated by the Ficoll gradient (L6115, Biochrom, Berlin, Germany) centrifugation of heparinized blood from 4 patients with LGI1 LE and 10 healthy volunteers. Cell separation was performed strictly according to a standardized and validated protocol using identical reagents from the same batches. Cells were washed once with serum-free AIM-V medium (Life Technologies, Darmstadt, Germany) followed by centrifugation at 400× *g* for 5 min.

### 4.4. Proliferation Assay

For the proliferation assays, freshly isolated PBMCs (10^5^ cells/100 µL) were incubated in quadruplicate cultures in 96-well microtiter culture plates (TPP, Trasadingen, Switzerland) in a serum-free AIM-V medium with 10 µg/mL of recombinant native LGI1, 10 µg/mL of denatured LGI1, or a concentrated control supernatant of non-transfected HEK 293 cell cultures. The cultures were incubated for 6 days at 37 °C in an incubator equipped with an IncuCyte S3^®^ live cell analysis system (Essen BioScience Ltd., Roysten, UK) to determine the cell proliferation. Four images per well from four wells were taken every 6 h after an initial 12 h resting period. Microscopic pictures were analyzed from day 4 with the IncuCyte^®^ S3 software (version IncuCyte 2021A) by counting cells bigger than 150 µm^2^ to exclude cell debris or apoptotic bodies to measure the LGI1-induced proliferation. The PBMC proliferation in LE patients and healthy volunteers was calculated as the cell count per image or as the stimulation index (SI), the cell count of LGI1-stimulated PBMCs divided by the cell count of untreated PBMCs in the medium).

### 4.5. Cytokine ELISA

For the measurement of cytokines, PBMCs (10^6^ cells/mL) were cultured in 24-well plates in an AIM-V medium with 10 µg/mL of native or denatured LGI1 or a concentrated control supernatant of non-transfected HEK 293 cell cultures. Due to limited amounts of PBMCs from patients with LGI1 ab-mediated LE, we quantified the cytokine concentrations only on day 4. After 4 days of incubation, the cell culture supernatants were harvested by centrifugation at 3000× *g* for 2 min and were stored immediately at −20 °C with temperature monitoring until the cytokine measurements. The concentrations of IFN-γ (#DIF50C), IL-5 (#D5000B), IL-10 (#D1000B), and IL-17 (#D1700) in these cell culture supernatants were determined using specific ELISA systems (bio-techne, R&D Systems, Minneapolis, MN, USA) according to the manufacturer’s instructions. Samples were applied undiluted, except for IL-10 with a 1:10 predilution.

### 4.6. Statistical Analysis

The statistical significances of the cell proliferation and cytokine production assays between the control and LGI1 LE were calculated with the Mann–Whitney test using GraphPad Prism software (version 7.05).

## 5. Conclusions

Although the number of patients included in the present study was very low, we hypothesize that in addition to antibody-mediated mechanisms, LGI1 AE autoreactive T cells might represent another possible pathomechanism for this disease. This may explain two clinical findings as being detached from the ab effect: (i) the low ab titers, in general, found in LGI-1 AE that were contradictory with the CNS damage occurring, and (ii) that although sometimes no LGI1 abs were found in the CSF, encephalitis occurred. With these data in mind, it is intriguing to speculate that T cell-mediated immunoreactions may play an additional role in the pathomechanisms of LGI1 AE. These putative T cell pathomechanisms may also, probably, be taken as pars pro toto for other AEs. Further studies with larger numbers of cases are necessary to further validate the observations presented.

## Figures and Tables

**Figure 1 ijms-25-02581-f001:**
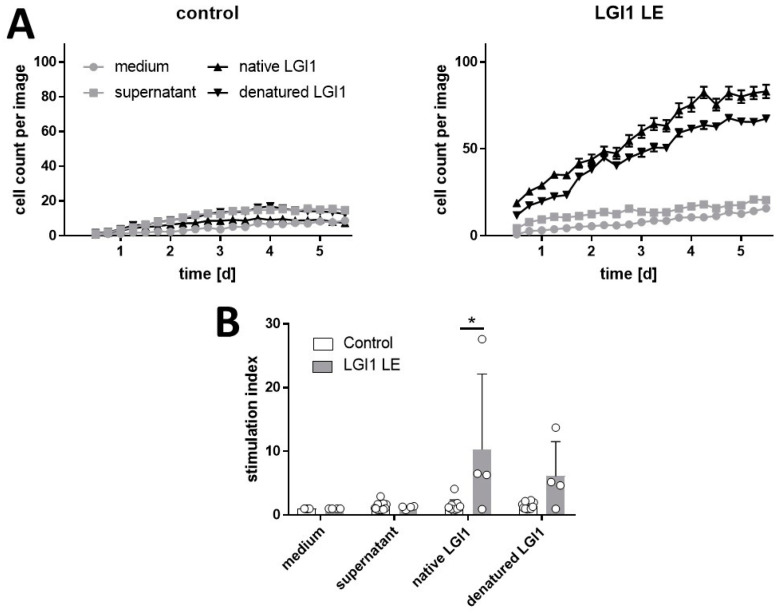
Native and denatured LGI1 stimulate the proliferation of PBMCs from patients with LGI1 ab-mediated LE but not those from healthy controls. PBMCs were incubated with recombinant native and denatured LGI1 or cell culture medium and supernatant of non-transfected HEK 293 cells as vehicle controls. Every 6 h, images of the cultures were taken to determine the cell proliferation rate using the IncuCyte S3^®^ live cell analysis system (Essen BioScience Ltd., Roysten, UK). (**A**) Representative time course of LGI1-induced cell proliferation during an incubation period of 5 days presented as cell count per image. (**B**) Cell proliferation at day 4 of independent experiments with 4 LGI1 LE patients and 10 healthy control volunteers presented as stimulation index (mean + SD; * *p* < 0.05). Raw data are shown in Appendix A.

**Figure 2 ijms-25-02581-f002:**
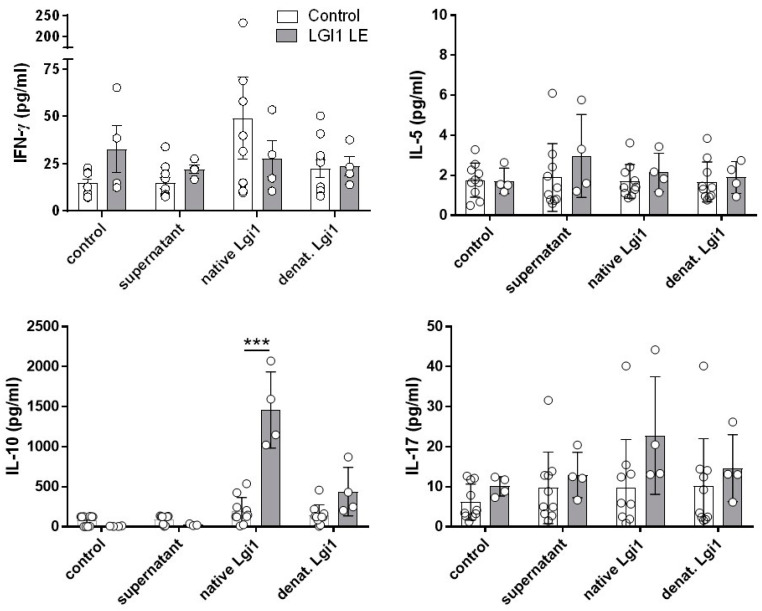
Native and denatured LGI1 induce production and secretion of IL-10 in PBMCs from patients with LGI1 LE but not those from healthy controls. PBMCs of 4 patients with LGI1 LE (grey bars) and 10 healthy control volunteers (white bars) were incubated with native or denatured recombinant LGI1 or with cell culture medium and supernatant of non-transfected HEK 293 cells as vehicle controls. Cell culture supernatants were harvested after 4 days, and cytokine concentrations were determined with specific ELISA. IFN-γ, IL-5, IL-10, and IL-17 production are shown as means + SD; *** *p* < 0.001. Raw data are shown in Appendix A.

## Data Availability

The data are contained within this article.

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
