# Peer review of "Leucine-Rich Glioma-Inactivated 1 (LGI1) Protein Stimulates Proliferation and IL-10 Production in Peripheral Blood Mononuclear Cells of Patients with LGI1 Antibody-Mediated Autoimmune Encephalitis In Vitro"

_ijms, 2024, doi:10.3390/ijms25052581_

Round 1

Reviewer 1 Report

Comments and Suggestions for Authors

The resubmitted paper entitled “LGI1 protein stimulates proliferation and IL-10 production in PBMC of patients with LGI1 antibody-mediated autoimmune encephalitis in vitro” is much improved. The authors expanded the control group of healthy participants from four to ten, unfortunately leaving the number of patients with LGI1 LE unchanged. Additional details regarding concentrations of cytokines are currently provided in Tables S1-S4. It may be useful to include sex and age data for both patients with LGI1 LE and healthy control participants in Tables S1-S4. The author should also explain the following data, why concentrations of IL-10 were very low in cell culture supernatants of PBMCs incubated with the medium or control supernatants of non-transfected HEK 293 cells from four (controls 1-4) of ten healthy individuals (Table S3, first and second columns). Since the authors reported data for both sexes, I also suggest to discuss sex differences in cytokines, particularly IL-10.

Minor

Line 17 – Anti should be “anti”

Line 34 - Autoimmune Encephalitis should be “Autoimmune encephalitis”

Lines 47, 48 - N-methyl-d-aspartate-receptors should be “N-methyl-D-aspartate receptors”

Comments on the Quality of English Language

Minor editing of English language required

Author Response

The resubmitted paper entitled “LGI1 protein stimulates proliferation and IL-10 production in PBMC of patients with LGI1 antibody-mediated autoimmune encephalitis in vitro” is much improved. The authors expanded the control group of healthy participants from four to ten, unfortunately leaving the number of patients with LGI1 LE unchanged. Additional details regarding concentrations of cytokines are currently provided in Tables S1-S4.

It may be useful to include sex and age data for both patients with LGI1 LE and healthy control participants in Tables S1-S4.

Response: We agree with the reviewer and included sex and age data for both patients with LGI1 LE and healthy control participants in a new table S1 of the „Supplement“ section.

The author should also explain the following data, why concentrations of IL-10 were very low in cell culture supernatants of PBMCs incubated with the medium or control supernatants of non-transfected HEK 293 cells from four (controls 1-4) of ten healthy individuals (Table S3, first and second columns).

Response: Since the stimulation experiments for individual healthy control volunteers took place at different times and seasons with different infection rates, differences in baseline IL-10 activity in supernatants of PBMCs incubated with medium or control supernatants of non-transfected HEK 293 cells are possible.

Since the authors reported data for both sexes, I also suggest to discuss sex differences in cytokines, particularly IL-10.

Response: This is a good idea but I think that our sample size is too small to draw any conclusion on this topic. Furthermore, when looking at other studies such as in our references 10 or 14, no sex differences have been reported. We would like to leave this question to larger studies having this question as a scope.

Minor

Line 17 – Anti should be “anti”

Line 34 - Autoimmune Encephalitis should be “Autoimmune encephalitis”

Lines 47, 48 - N-methyl-d-aspartate-receptors should be “N-methyl-D-aspartate receptors”

Response: We thank the reviewer and corrected these three notes.

Reviewer 2 Report

Comments and Suggestions for Authors

Goihl et al. described the effects of IGI1 on IL10 cytokine production from the ex vivo PBMC of patients with LE. Kindly refer to my comments: 

1) Title: Please avoid abbreviations

2) Abstract (line 17): LGI1 abbreviation should be put before the antibodies 

3) Abstract: the research gap is not clear. 

4) Introduction (line): IgG4? The authors shall introduce it first. Why focus on IgG4 not other subtypes? 

5)  Kindly follow the format of journal on the subheading sequences (Methods and Materials first)

6) Methods (line 153): Kindly provide the sources of cells. Please provide the cell line authentication as mentioned in the journal guidelines if possible. 

7) Methods (line 172): Be specific on the local ethics committee. Please state the date of approval. 

8) Methods (line 174-190): Kindly mention the date of hospitalization /administration. 

9) Methods (line 195): the catalogue number of Ficoll is missing. 

10) Methods (line 198-206): Did the authors leave the PBMC to be stabilise (equilibrium) for overnight or 24h before the IGI1 experiment? 

11) Methods (line 77-78 & 204): IncuCyte itself is not able to differentiate the viability of cells without staining. The counted cells are including the dead cells. How about the cell debris and/or apoptotic bodies? 

12) Methods (line 216-217): Kindly include the catalog number of ELISA kit. Any predilution on samples (supernatant) before the ELISA? 

13) Statistical analysis (line 218-221): are the data normally distributed? what is the significant level? 

14) Figure 1A: Kindly use grey color for 2 of the plot (medium and supernatant). What do you mean by "cell count per image"? How many images were taken for measurement?

15) Figure 1B: I am doubt with the reported significant difference as you can see the SEM is very large and even larger than the mean of samples (Native IGI1). Kindly attach the raw data and the statistical analysis. 

16) Figure 2: Be standardize with the label (IGI1 vs Igi1)

17) Figure 2: Why measuring this at only day 4? You are not able to capture the optimal changes. 

18) Discussion (line 112): "B cell answer elicits a concomitant T cell answer"? 

19) Discussion: Please discuss more on IL10. As far as i know, IL10 is more towards as an anti-inflammatory cytokine. How this relates with IGI1 and the AE pathophtysiology? 

20) Discussion: Please include the limitations of your study. Please have a separate paragraphor subheading for conclusion

Comments on the Quality of English Language

It is generally acceptable. 

Author Response

  • Title: Please avoid abbreviations

Response: We changed the titel to: Leucine-rich glioma inactivated 1 (LGI1) protein stimulates proliferation and IL-10 production in peripheral blood mononuclear cells of patients with LGI1 antibody-mediated auto-immune encephalitis in vitro

  • Abstract (line 17): LGI1 abbreviation should be put before the antibodies 

Response: We corrected this.

  • Abstract: the research gap is not clear. 

Response: We wanted to look at whether t-cells are also involved in the pathomechanisms of this disease. To stress this point better we changed the sentence to “However, it is unclear, whether an additional disease-related LGI1 antigen-specific T cell response is involved in the pathogenesis of this disease.“ We hope that this change will better explain the research gap.

  • Introduction (line): IgG4? The authors shall introduce it first. Why focus on IgG4 not other subtypes? 

Response: We added some more sentences explaining and introducing the IgG4 related diseases in the introduction

  • Kindly follow the format of journal on the subheading sequences (Methods and Materials first)

Response: We included a „Conclusion“ paragraph at the end of the manuscript. We again also used the template provided by the journal itself.

  • Methods (line 153): Kindly provide the sources of cells. Please provide the cell line authentication as mentioned in the journal guidelines if possible. 

Response: We included these information in the „Materials and Methods“ section.

  • Methods (line 172): Be specific on the local ethics committee. Please state the date of approvai.

Response: We included this information.

  • Methods (line 174-190): Kindly mention the date of hospitalization /administration. 

Response: We could not mention the precise date of the admission for each patient due to data protection issues but provided the years of the start and the years the blood was took for this study. Furthermore patients were in the out-patient clinic when they gave their blood. We stated a more general information about 2019 as year the blood was taken. We do think that this information is enough and would like to avoid a query as this fact does not seem as a major obstacle for this excellent study.

  • Methods (line 195): the catalogue number of Ficoll is missing. 

Response: We added this information in the „Materials and Methods“ section.

10) Methods (line 198-206): Did the authors leave the PBMC to be stabilise (equilibrium) for overnight or 24h before the IGI1 experiment? 

Response: We used the freshly isolated PBMC for all experiments. We added this information in the „Materials and Methods“ section.

11) Methods (line 77-78 & 204): IncuCyte itself is not able to differentiate the viability of cells without staining. The counted cells are including the dead cells. How about the cell debris and/or apoptotic bodies? 

Response: We thank the reviewer for this comment. We changed the information in the „Materials and Methods“ section: „Microscopic pictures were analyzed from day 4 with the IncuCyte® S3 software by counting cells bigger than 150 µm² to exclude cell debris or apoptotic bodies to measure the LGI1-induced proliferation.“

12) Methods (line 216-217): Kindly include the catalog number of ELISA kit. Any predilution on samples (supernatant) before the ELISA? 

Response: We included these information in the „Materials and Methods“ section.

13) Statistical analysis (line 218-221): are the data normally distributed? what is the significant level? 

Response: We thank the reviewer for this comment. We re-evaluated the statistical analysis using the Mann-Whitney-test and included these information in the „Materials and Methods“ section: „Statistical significances of cell proliferation and cytokine production assays between control and LGI1 LE were calculated with Mann-Whitney-test using GraphPad Prism software.“

14) Figure 1A:  Kindly use grey color for 2 of the plot (medium and supernatant). What do you mean by "cell count per image"? How many images were taken for measurement?

Response: We changed the color in Figure 1A as suggested by the reviewer.

One image of the entire well of the titer plate was included in the analysis.  We changed this information in the „Materials and Methods“ section: „PBMC proliferation from LE patients and healthy volunteers was calculated as cell count per image or as stimulation index (SI, cell count of LGI1-stimulated PBMC divided by cell count of untreated PBMC in medium).“

15) Figure 1B: I am doubt with the reported significant difference as you can see the SEM is very large and even larger than the mean of samples (Native IGI1). Kindly attach the raw data and the statistical analysis. 

Response: We thank the reviewer for this comment. We re-evaluated the data of our two figures. Indeed the data of both Figures are presented as SD and not as SEM. We changed this error in the legends of figures 1 and 2. Moreover, we re-calculated the statistical analysis using the Mann-Whitney-test (see13). The raw data of the proliferation experiments are now presented as new table S2.

16) Figure 2: Be standardize with the label (IGI1 vs Igi1)

Response: We standardized the label LGI1 in figure 2.

17) Figure 2: Why measuring this at only day 4? You are not able to capture the optimal changes. 

Response: Due to limited amounts of PBMC from the LGI1-LE patients, we had to limit the experiments to one time point for cytokine analysis. Based on experience in cytokine measurement in culture supernatants of antigen-stimulated PBMC, we decided on the time point of 4 days.

18) Discussion (line 112): "B cell answer elicits a concomitant T cell answer"? 

Response: We thank the reviewer for this comment. We changed the sentence to:

„Since a B cell response is often triggered by a T cell response“.

19) Discussion: Please discuss more on IL10. As far as i know, IL10 is more towards as an anti-inflammatory cytokine. How this relates with IGI1 and the AE pathophtysiology? 

Response: We thank the reviewer for this suggestion and modified the IL-10 paragraph in the „Discussion“.    We included the information that „Alongside TGF-ß, IL-10 is one of the most potent immunosuppressive factors, which inhibits T-cell activation in particular. “ As we stated in the manuscript that IgG4-related diseases and Il-10 could be closer to each other than thought before due to the role of Il-10 in the IgG class switch.

20) Discussion: Please include the limitations of your study.

Response: We put a paragraph between the discussion and conclusion addressing the limitations of our study.

21) Please have a separate paragraphor subheading for conclusion               

Response: We modified the last paragraph of the discussion into an independent  „Conclusion“ paragraph at the end of the manuscript.

Round 2

Reviewer 2 Report

Comments and Suggestions for Authors

Kindly refer to the follow-up comments: 

  • Abstract: the research gap is not clear. 

Response: We wanted to look at whether t-cells are also involved in the pathomechanisms of this disease. To stress this point better we changed the sentence to “However, it is unclear, whether an additional disease-related LGI1 antigen-specific T cell response is involved in the pathogenesis of this disease.“ We hope that this change will better explain the research gap.

Follow-up comment: It looks good for your Abstract. Kindly include necessary info in the Introduction, for instance, what we know about T cells in the pathogenesis of AE, how this links up with IgG4 , etc. This is crucial to emphasize your research gap. 

  • Introduction (line): IgG4? The authors shall introduce it first. Why focus on IgG4 not other subtypes? 

Response: We added some more sentences explaining and introducing the IgG4 related diseases in the introduction.

Follow-up comment: Please define IgG4 in full. Good for readers that new in this field. 

  • Methods (line 172): Be specific on the local ethics committee. Please state the date of approvai.

Response: We included this information.

Follow-up comment: Kindly check for typing errors. Were the authors carried out the study according to the Declaration of Helsinki 1975? Your previous study (REF 22) was published with the same study number (100/16). Are these 2 studies using the same ethical approval? 

  • Methods (line 174-190): Kindly mention the date of hospitalization /administration. 

Response: We could not mention the precise date of the admission for each patient due to data protection issues but provided the years of the start and the years the blood was took for this study. Furthermore patients were in the out-patient clinic when they gave their blood. We stated a more general information about 2019 as year the blood was taken. We do think that this information is enough and would like to avoid a query as this fact does not seem as a major obstacle for this excellent study.

Follow-up comment: Were all controls and patients being drawn the blood at the same time or day? How to ensure the consistency of data if they are not conducted in the same time. 

  • Methods (line 195): the catalogue number of Ficoll is missing. 

Response: We added this information in the „Materials and Methods“ section.

Follow-up comment: The Ficoll 1.077 is used to isolate PBMC which includes B cells, T cells,  monocytes, NK cells (including NKT) and dendritic cells. Kindly note that all these cells can produce IFN, IL-5,  IL-10 and IL-17. How do the authors rule out these possibilities and conclude this cytokine profile is solely the T cells? 

11) Methods (line 77-78 & 204): IncuCyte itself is not able to differentiate the viability of cells without staining. The counted cells are including the dead cells. How about the cell debris and/or apoptotic bodies? 

Response: We thank the reviewer for this comment. We changed the information in the „Materials and Methods“ section: „Microscopic pictures were analyzed from day 4 with the IncuCyte® S3 software by counting cells bigger than 150 µm² to exclude cell debris or apoptotic bodies to measure the LGI1-induced proliferation.“

Follow-up comment: Cell viability dye is very common, affordable and reliable. I do not see any limitation in terms of instrument compatibility. Therefore,  it is not acceptable not to use them to determine the proliferation of cells. Kindly include the images from IncuCyte S3 and attach them as supplementary data to support your interpretation. 

12) Methods (line 216-217): Kindly include the catalog number of ELISA kit. Any predilution on samples (supernatant) before the ELISA? 

Response: We included these information in the „Materials and Methods“ section.

Follow-up comment: Please state the manufacturer of the ELISAs. Please check the typing error for IL-17. 

13) Statistical analysis (line 218-221): are the data normally distributed? what is the significant level? 

Response: We thank the reviewer for this comment. We re-evaluated the statistical analysis using the Mann-Whitney-test and included these information in the „Materials and Methods“ section: „Statistical significances of cell proliferation and cytokine production assays between control and LGI1 LE were calculated with Mann-Whitney-test using GraphPad Prism software.“

14) Figure 1A:  Kindly use grey color for 2 of the plot (medium and supernatant). What do you mean by "cell count per image"? How many images were taken for measurement?

Response: We changed the color in Figure 1A as suggested by the reviewer.

One image of the entire well of the titer plate was included in the analysis.  We changed this information in the „Materials and Methods“ section: „PBMC proliferation from LE patients and healthy volunteers was calculated as cell count per image or as stimulation index (SI, cell count of LGI1-stimulated PBMC divided by cell count of untreated PBMC in medium).“

Follow-up comment: Thank you the authors. Due to the nature of blood cells, they will tend to (temporarily or weakly) attach or sediment over the edge or centre of the well. Therefore, single-image analysis is not acceptable due to the high risk of reporting bias and error. Please repeat the experiments or include the images here to convince the readers. 

15) Figure 1B: I am doubt with the reported significant difference as you can see the SEM is very large and even larger than the mean of samples (Native IGI1). Kindly attach the raw data and the statistical analysis. 

Response: We thank the reviewer for this comment. We re-evaluated the data of our two figures. Indeed the data of both Figures are presented as SD and not as SEM. We changed this error in the legends of figures 1 and 2. Moreover, we re-calculated the statistical analysis using the Mann-Whitney-test (see13). The raw data of the proliferation experiments are now presented as new table S2.

Follow-up comment: Thank you the authors for the raw data. There are some extremely small and big data for those groups with big error bars. It is more sensitive when your sample size is small. Is there any technical replicate for Table S3, S4, S5 and S6? 

16) Figure 2: Be standardize with the label (IGI1 vs Igi1)

Response: We standardized the label LGI1 in figure 2.

Follow-up comment: Kindly label the bars for Figure 1B and 2, not the dot color. Besides, please use the dual-scaled Y axis for IL-10 to accommodate the extreme values. 

17) Figure 2: Why measuring this at only day 4? You are not able to capture the optimal changes. 

Response: Due to limited amounts of PBMC from the LGI1-LE patients, we had to limit the experiments to one time point for cytokine analysis. Based on experience in cytokine measurement in culture supernatants of antigen-stimulated PBMC, we decided on the time point of 4 days.

Follow-up comment: This is arguable and kindly clarify this in the manuscript. Please support with literature/previous studies

19) Discussion: Please discuss more on IL10. As far as i know, IL10 is more towards as an anti-inflammatory cytokine. How this relates with IGI1 and the AE pathophtysiology? 

Response: We thank the reviewer for this suggestion and modified the IL-10 paragraph in the „Discussion“.    We included the information that „Alongside TGF-ß, IL-10 is one of the most potent immunosuppressive factors, which inhibits T-cell activation in particular. “ As we stated in the manuscript that IgG4-related diseases and Il-10 could be closer to each other than thought before due to the role of Il-10 in the IgG class switch.

Follow-up comment: Please amend this "IL-10 clearly promotes B cell activation" because IL-10 also suppresses the B cells. Please support with citations.

20) Discussion: Please include the limitations of your study.

Response: We put a paragraph between the discussion and conclusion addressing the limitations of our study.

Follow-up comment: You have more limitations (technical) that may need to be discussed as well. 

21) Please have a separate paragraphor subheading for conclusion               

Response: We modified the last paragraph of the discussion into an independent  „Conclusion“ paragraph at the end of the manuscript.

Follow-up comment: Do not forget to conclude your data and incorporate it into the conclusion. Avoid overclaiming or too much speculation without valid data as support. Kindly amend the conclusion accordingly

Author Response

Submission of Revised Manuscript # ijms-2769094 to “International Journal of Molecular Sciences“

Point-by-point reply

We thank the reviewer for the comments and evaluation of our manuscript. We agree that quality of the manuscript has improved as a result of the changes requested by the reviewer.

Reviewer 2:

  • Abstract: the research gap is not clear.

Response: We wanted to look at whether T-cells are also involved in the pathomechanisms of this disease. To stress this point better we changed the sentence to “However, it is unclear, whether an additional disease-related LGI1 antigen-specific T cell response is involved in the pathogenesis of this disease.“ We hope that this change will better explain the research gap.

Follow-up comment: It looks good for your Abstract. Kindly include necessary info in the Introduction, for instance, what we know about T cells in the pathogenesis of AE, how this links up with IgG4 , etc. This is crucial to emphasize your research gap.

Follow-up response: We modified the “gap message” in the introduction to:

„However, it is unclear, whether a disease-related LGI1 antigen specific effector T cell response occurs in this disease with subsequent cytokine release by disease-related, antigen-specific immune cells, which is involved in the pathogenesis of LGI1 LE.“

  • Introduction (line): IgG4? The authors shall introduce it first. Why focus on IgG4 not other subtypes?

Response: We added some more sentences explaining and introducing the IgG4 related diseases in the introduction.

Follow-up comment: Please define IgG4 in full. Good for readers that new in this field.

Follow-up response: dear reviewer, we already included a lot information about IgG4 following your comments. Please see lines 46-52:

„This notion received further support, when identifying that LGI1 AE might be an IgG4-mediated disease characterized by a lack of significant neuroimmunological response as measured in CSF and serum. IgG4-related diseases such as Musk-antibody mediated myasthenia gravis or mikulicz´s diseases are a specific class of immunological diseases having a predominant role of IgG4 antibodies in their pathomechanisms. These IgG-class 4 antibodies are poor complement-binders and are different in their clinical picture than other IgG-related/mediated diseases such as other autoimmune encephalitis.”

We already explained IgG4, the IgG4-related diseases and it´s connection to LGI1. We do think that this strong enough for the readers interested in this manuscript.

(3) Methods (line 172): Be specific on the local ethics committee. Please state the date of approvai.

Response: We included this information.

Follow-up comment: Kindly check for typing errors. Were the authors carried out the study according to the Declaration of Helsinki 1975? Your previous study (REF 22) was published with the same study number (100/16). Are these 2 studies using the same ethical approval?

Follow-up response:

Yes, this is the general ethical approval for autoimmune encephalitis patients from Magdeburg.

(4) Methods (line 174-190): Kindly mention the date of hospitalization /administration.

Response: We could not mention the precise date of the admission for each patient due to data protection issues but provided the years of the start and the years the blood was took for this study. Furthermore patients were in the out-patient clinic when they gave their blood. We stated a more general information about 2019 as year the blood was taken. We do think that this information is enough and would like to avoid a query as this fact does not seem as a major obstacle for this excellent study.

Follow-up comment: Were all controls and patients being drawn the blood at the same time or day? How to ensure the consistency of data if they are not conducted in the same time.

Follow-up response: Dear reviewer, we do not understand this point. The consistency of the data is not time-based but method based, as we carried out precisely the same experiment every time. As this disease is very rare there is absolutely no way to do this investigations at the same time in all patients.

Our 10 controls were done in three batches to easy up things. But again, we could have performed them at 10 different days due to the high standardization of the experiment without loss of consistency. If this would be different when using an in-house assay etc. for experiments. In summary, we do not have batch effects here.

(5) Methods (line 195): the catalogue number of Ficoll is missing.

Response: We added this information in the „Materials and Methods“ section.

Follow-up comment: The Ficoll 1.077 is used to isolate PBMC which includes B cells, T cells,  monocytes, NK cells (including NKT) and dendritic cells. Kindly note that all these cells can produce IFN, IL-5,  IL-10 and IL-17. How do the authors rule out these possibilities and conclude this cytokine profile is solely the T cells?

Follow-up response:

We thank the reviewer for this comment.

In the present version auf the manuscript, we wrote as yet in the title, the abstract, the "results" section that we studied cytokine production of PBMC.

We included the information that PBMC are T cells, B cells, NK cells, monocytes and dendritic cells in the “Discussion” section:

“It should be noted that PBMC are a cell mixture of PBMC are T cells, B cells, NK cells, monocytes and dendritic cells, which all produce cytokines. In the present experimental setting we can not clearly say, which cells produce which amount of IL‑10. It is undisputed that IL-10 is produced also by T cells following such stimulation with LGI1 antigen.”

Moreover, we pointed out as a limitation of the study that the present investigations were not additionally carried out on purified T cells with irradiated feeder cells.

11) Methods (line 77-78 & 204): IncuCyte itself is not able to differentiate the viability of cells without staining. The counted cells are including the dead cells. How about the cell debris and/or apoptotic bodies?

Response: We thank the reviewer for this comment. We changed the information in the „Materials and Methods“ section: „Microscopic pictures were analyzed from day 4 with the IncuCyte® S3 software by counting cells bigger than 150 µm² to exclude cell debris or apoptotic bodies to measure the LGI1-induced proliferation.“

Follow-up comment: Cell viability dye is very common, affordable and reliable. I do not see any limitation in terms of instrument compatibility. Therefore,  it is not acceptable not to use them to determine the proliferation of cells. Kindly include the images from IncuCyte S3 and attach them as supplementary data to support your interpretation.

Follow-up response:

Dear reviewer, thank you very much for this important point. We out the original data into the supplement as Figure S1.

12) Methods (line 216-217): Kindly include the catalog number of ELISA kit. Any predilution on samples (supernatant) before the ELISA?

Response: We included these information in the „Materials and Methods“ section.

Follow-up comment: Please state the manufacturer of the ELISAs. Please check the typing error for IL-17.

Follow-up response:

We included the information and corrected the typing error.

14) Figure 1A:  Kindly use grey color for 2 of the plot (medium and supernatant). What do you mean by "cell count per image"? How many images were taken for measurement?

Response: We changed the color in Figure 1A as suggested by the reviewer.

One image of the entire well of the titer plate was included in the analysis.  We changed this information in the „Materials and Methods“ section: „PBMC proliferation from LE patients and healthy volunteers was calculated as cell count per image or as stimulation index (SI, cell count of LGI1-stimulated PBMC divided by cell count of untreated PBMC in medium).“

Follow-up comment: Thank you the authors. Due to the nature of blood cells, they will tend to (temporarily or weakly) attach or sediment over the edge or centre of the well. Therefore, single-image analysis is not acceptable due to the high risk of reporting bias and error. Please repeat the experiments or include the images here to convince the readers.

Follow-up response:

We included the images in figure S1 of the supplement. As one can see the images are sufficient to draw our conclusion. We cannot repeat the experiments due to fact that our LGI1 patients recovered.

Moreover, we included into the “Method” section the sentence:

“Four images per well from four wells were taken every 6 h after an initial 12 h resting period.”

15) Figure 1B: I am doubt with the reported significant difference as you can see the SEM is very large and even larger than the mean of samples (Native IGI1). Kindly attach the raw data and the statistical analysis.

Response: We thank the reviewer for this comment. We re-evaluated the data of our two figures. Indeed the data of both Figures are presented as SD and not as SEM. We changed this error in the legends of figures 1 and 2. Moreover, we re-calculated the statistical analysis using the Mann-Whitney-test (see13). The raw data of the proliferation experiments are now presented as new table S2.

Follow-up comment: Thank you the authors for the raw data. There are some extremely small and big data for those groups with big error bars. It is more sensitive when your sample size is small. Is there any technical replicate for Table S3, S4, S5 and S6?

Follow-up response:

There are no technical replicats in the data of Tables S3 – S6.

16) Figure 2: Be standardize with the label (IGI1 vs Igi1)

Response: We standardized the label LGI1 in figure 2.

Follow-up comment: Kindly label the bars for Figure 1B and 2, not the dot color. Besides, please use the dual-scaled Y axis for IL-10 to accommodate the extreme values.

Follow-up response:

We thank the reviewer for this comment. We changed the figures 1B and 2, respectively.

17) Figure 2: Why measuring this at only day 4? You are not able to capture the optimal changes.

Response: Due to limited amounts of PBMC from the LGI1-LE patients, we had to limit the experiments to one time point for cytokine analysis. Based on experience in cytokine measurement in culture supernatants of antigen-stimulated PBMC, we decided on the time point of 4 days.

Follow-up comment: This is arguable and kindly clarify this in the manuscript. Please support with literature/previous studies

Follow-up response:

We included this information in the “Methods” section and pointed the fact as limitation and suggestion for future studies in the “Discussion”.

19) Discussion: Please discuss more on IL10. As far as i know, IL10 is more towards as an anti-inflammatory cytokine. How this relates with IGI1 and the AE pathophtysiology?

Response: We thank the reviewer for this suggestion and modified the IL-10 paragraph in the „Discussion“.    We included the information that „Alongside TGF-ß, IL-10 is one of the most potent immunosuppressive factors, which inhibits T-cell activation in particular. “ As we stated in the manuscript that IgG4-related diseases and Il-10 could be closer to each other than thought before due to the role of Il-10 in the IgG class switch.

Follow-up comment: Please amend this "IL-10 clearly promotes B cell activation" because IL-10 also suppresses the B cells. Please support with citations.

Follow-up response:

We thank the reviewer for this hint and added a sentence in the discussion: Regarding B-cells, IL-10 can also suppress B-cells in certain pathological condition with the reference. We have to state that IL-10 is not either a B-cell activator or B-cell suppressor.

After studying the literature it looks like IL-10 changes roles in diseases. Therefore, we also added a sentence concerning this later in the discussion: “The precise effect from IL-10 seems to be determined by the underlying disease.”

20) Discussion: Please include the limitations of your study.

Response: We put a paragraph between the discussion and conclusion addressing the limitations of our study.

Follow-up comment: You have more limitations (technical) that may need to be discussed as well.

Follow-up response:

We included the following sentences in the „Discussion“ section:

„Due to limited amounts of PBMC from patients with LGI1 ab-mediated LE, we quanti-fied the cytokine concentrations in PBMC culture supernatants only on day 4 of the experiment. In future studies, this should be done at several time points. Moreover, another limitation of the present study is that the stimulation experiments with LGI1 were not additionally carried out on purified T cells, e.g. with irradiated feeder cells as antigen presenting cells.“

21) Please have a separate paragraphor subheading for conclusion               

Response: We modified the last paragraph of the discussion into an independent „Conclusion“ paragraph at the end of the manuscript.

Follow-up comment: Do not forget to conclude your data and incorporate it into the conclusion. Avoid overclaiming or too much speculation without valid data as support. Kindly amend the conclusion accordingly

Follow-up response:

We think that we have concluded our data at the beginning of the discussion section with:

“We observed that recombinant LGI1 protein stimulated the proliferation of PBMCs from patients with LGI1 ab-mediated LE in vitro, but not of PBMC from healthy control volunteers. Since a B cell response is often triggered by a T cell response, we therefore started to examine the T cell answer in autoimmune encephalitis. This result demonstrates the possible existence of autoreactive pathogenic T cells in these patients, which has not been reported previously in any known AE so far. “

We also think that we can use a little bit of hypothetical interpretation at this point. Although we followed your point and attenuated the manuscript from:

“It is now of upmost importance in order to avoid more CNS damage to also take into account possible T cell-mediated effects in LGI1 AE.“ to: “With this data in mind it is intriguing to speculate that T-cell mediated immunoreactions may play an additional role in the pathomechanisms of the LGI1 AE.”

Round 3

Reviewer 2 Report

Comments and Suggestions for Authors

Thank you the authors, kindly refer to some minor comments: 

(1) Introduction (line): IgG4? The authors shall introduce it first. Why focus on IgG4 not other subtypes?

Response: We added some more sentences explaining and introducing the IgG4 related diseases in the introduction.

Follow-up comment: Please define IgG4 in full. Good for readers that new in this field.

Follow-up response: dear reviewer, we already included a lot information about IgG4 following your comments. Please see lines 46-52:

Follow-up comments: Thank you the authors. You have misunderstood my earlier comments. Kindly DEFINE IgG4 as Immunoglobulin G4. 

(4) Methods (line 174-190): Kindly mention the date of hospitalization /administration.

Response: We could not mention the precise date of the admission for each patient due to data protection issues but provided the years of the start and the years the blood was took for this study. Furthermore patients were in the out-patient clinic when they gave their blood. We stated a more general information about 2019 as year the blood was taken. We do think that this information is enough and would like to avoid a query as this fact does not seem as a major obstacle for this excellent study.

Follow-up comment: Were all controls and patients being drawn the blood at the same time or day? How to ensure the consistency of data if they are not conducted in the same time.

Follow-up response: Dear reviewer, we do not understand this point. The consistency of the data is not time-based but method based, as we carried out precisely the same experiment every time. As this disease is very rare there is absolutely no way to do this investigations at the same time in all patients.

Our 10 controls were done in three batches to easy up things. But again, we could have performed them at 10 different days due to the high standardization of the experiment without loss of consistency. If this would be different when using an in-house assay etc. for experiments. In summary, we do not have batch effects here.

Follow-up comments: Thank you the authors. Ideally, we do not want to have any batch effect here, but you need to show to us (and readers) how you avoid the potential batch effect and ensure consistency in your study. Kindly add in explanations in your manuscript on the cell isolation (about the 10 different days) and how you ensure there is no batch effect in cytokine measurement (for example proper storage of culture supernatant). 

I have no further comment and thank you for the correction. 

Author Response

Thank you the authors, kindly refer to some minor comments:

(1) Introduction (line): IgG4? The authors shall introduce it first. Why focus on IgG4 not other subtypes?

Response: We added some more sentences explaining and introducing the IgG4 related diseases in the introduction.

Follow-up comment: Please define IgG4 in full. Good for readers that new in this field.

Follow-up response: dear reviewer, we already included a lot information about IgG4 following your comments. Please see lines 46-52:

Follow-up comments: Thank you the authors. You have misunderstood my earlier comments. Kindly DEFINE IgG4 as Immunoglobulin G4.

Follow-up response: We included „Immunoglobulin G4 (IgG4) in the „Introduction“.

(4) Methods (line 174-190): Kindly mention the date of hospitalization /administration.

Response: We could not mention the precise date of the admission for each patient due to data protection issues but provided the years of the start and the years the blood was took for this study. Furthermore patients were in the out-patient clinic when they gave their blood. We stated a more general information about 2019 as year the blood was taken. We do think that this information is enough and would like to avoid a query as this fact does not seem as a major obstacle for this excellent study.

Follow-up comment: Were all controls and patients being drawn the blood at the same time or day? How to ensure the consistency of data if they are not conducted in the same time.

Follow-up response: Dear reviewer, we do not understand this point. The consistency of the data is not time-based but method based, as we carried out precisely the same experiment every time. As this disease is very rare there is absolutely no way to do this investigations at the same time in all patients.

Our 10 controls were done in three batches to easy up things. But again, we could have performed them at 10 different days due to the high standardization of the experiment without loss of consistency. If this would be different when using an in-house assay etc. for experiments. In summary, we do not have batch effects here.

Follow-up comments: Thank you the authors. Ideally, we do not want to have any batch effect here, but you need to show to us (and readers) how you avoid the potential batch effect and ensure consistency in your study. Kindly add in explanations in your manuscript on the cell isolation (about the 10 different days) and how you ensure there is no batch effect in cytokine measurement (for example proper storage of culture supernatant).

Follow-up response: We included the sentences „Cell separation was performed strictly according to a standardized and validated protocol using identical reagents from the same batches“ and „ … cell culture supernatants  … were stored immediately at -20°C with temperature monitoring until cytokine measurement“ in the „materials and methods“ section.
